# Characterization of an African Swine Fever Virus Field Isolate from Vietnam with Deletions in the Left Variable Multigene Family Region

**DOI:** 10.3390/v16040571

**Published:** 2024-04-07

**Authors:** Aruna Ambagala, Kalhari Goonewardene, Ian El Kanoa, Thi Tam Than, Van Tam Nguyen, Thi Ngoc Ha Lai, Thi Lan Nguyen, Cassidy N. G. Erdelyan, Erin Robert, Nikesh Tailor, Chukwunonso Onyilagha, Lindsey Lamboo, Katherine Handel, Michelle Nebroski, Oksana Vernygora, Oliver Lung, Van Phan Le

**Affiliations:** 1National Centre for Foreign Animal Disease, Canadian Food Inspection Agency, Winnipeg, MB R3E 3M4, Canada; kalhari.goonewardene@inspection.gc.ca (K.G.); ian.elkanoa@inspection.gc.ca (I.E.K.); cass.erdelyan@inspection.gc.ca (C.N.G.E.); erin.robert@inspection.gc.ca (E.R.); nikesh.tailor@phac-aspc.gc.ca (N.T.); chukwunonso.onyilagha@inspection.gc.ca (C.O.); lindsey.lamboo@phac-aspc.gc.ca (L.L.); katherine.handel@inspection.gc.ca (K.H.); michelle.nebroski@inspection.gc.ca (M.N.); oksana.vernygora@inspection.gc.ca (O.V.); oliver.lung@inpection.gc.ca (O.L.); 2Department of Medical Microbiology and Infectious Diseases, Max Rady College of Medicine, University of Manitoba, Winnipeg, MB R3E 0J9, Canada; 3Comparative Biology and Experimental Medicine, Faculty of Veterinary Medicine, University of Calgary, Calgary, AB T2N 1N4, Canada; 4College of Veterinary Medicine, Vietnam National University of Agriculture, Hanoi 100000, Vietnam; thantam207@gmail.com (T.T.T.); hoangocha1998@gmail.com (T.N.H.L.); nguyenlan@vnua.edu.vn (T.L.N.); 5Institute of Veterinary Science and Technology, Hanoi 100000, Vietnam; nguyentammicro@gmail.com

**Keywords:** African swine fever, live-attenuated virus, horizontal transmission, virulence, MGF, GUS

## Abstract

In this paper, we report the characterization of a genetically modified live-attenuated African swine fever virus (ASFV) field strain isolated from Vietnam. The isolate, ASFV-GUS-Vietnam, belongs to p72 genotype II, has six multi-gene family (MGF) genes deleted, and an *Escherichia coli* GusA gene (GUS) inserted. When six 6–8-week-old pigs were inoculated with ASFV-GUS-Vietnam oro-nasally (2 × 10^5^ TCID_50_/pig), they developed viremia, mild fever, lethargy, and inappetence, and shed the virus in their oral and nasal secretions and feces. One of the pigs developed severe clinical signs and was euthanized 12 days post-infection, while the remaining five pigs recovered. When ASFV-GUS-Vietnam was inoculated intramuscularly (2 × 10^3^ TCID_50_/pig) into four 6-8 weeks old pigs, they also developed viremia, mild fever, lethargy, inappetence, and shed the virus in their oral and nasal secretions and feces. Two contact pigs housed together with the four intramuscularly inoculated pigs, started to develop fever, viremia, loss of appetite, and lethargy 12 days post-contact, confirming horizontal transmission of ASFV-GUS-Vietnam. One of the contact pigs died of ASF on day 23 post-contact, while the other one recovered. The pigs that survived the exposure to ASFV-GUS-Vietnam via the mucosal or parenteral route were fully protected against the highly virulent ASFV Georgia 2007/1 challenge. This study showed that ASFV-GUS-Vietnam field isolate is able to induce complete protection in the majority of the pigs against highly virulent homologous ASFV challenge, but has the potential for horizontal transmission, and can be fatal in some animals. This study highlights the need for proper monitoring and surveillance when ASFV live-attenuated virus-based vaccines are used in the field for ASF control in endemic countries.

## 1. Introduction

African swine fever (ASF) is a contagious fatal hemorrhagic fever of domestic and wild pigs [1]. The causative agent, African swine fever virus (ASFV), is a highly stable, complex, large DNA virus belonging to the family *Asfaviridae* [2]. ASFV continues to spread in Africa, Europe, Russia, and Asia, causing high mortality in domestic and wild pigs and resulting in serious socio-economic consequences [3]. In 2021, ASFV spread to the Dominican Republic and Haiti, and became endemic in the island of Hispaniola, increasing the potential spread of ASF to the other countries in the Americas [4,5].

The current pandemic strain of ASFV belonging to the p72 genotype II was first detected in 2007 in Georgia [6] and it later spread to Armenia, Russia, and Eastern Europe [7]. In 2018, ASFV emerged in China, the country with the world’s largest pig population [8]. The virus spread across China within months and then to the neighboring countries including Vietnam, Cambodia, Laos, the Philippines, Malaysia, India, and most recently, Bangladesh, causing the world’s biggest animal disease outbreak ever reported [9,10]. As the virus spread and killed millions of pigs, scientists around the world intensified their efforts to develop an effective ASF vaccine [11]. As a result, a number of genetically modified ASF live-attenuated virus (LAV) strains have been generated as vaccine candidates in China, Europe, and the USA [12,13,14,15]. In addition to genetically modified strains, naturally attenuated ASFV strains have been proposed as potential vaccine candidates [16]. Two of the genetically modified p72 genotype II LAV strains, have been licensed in Vietnam. Recently, Vietnam, the Philippines, and the Dominican Republic, three ASF endemic countries, have conducted field evaluations of these LAV vaccines [17,18,19]. None of the LAV vaccines have serological DIVA capabilities, and therefore, vaccinated animals cannot be serologically differentiated from naturally infected animals.

To date, no LAV vaccines are licensed in China. However, since 2019, a number of LAV strains have been detected in pig farms in China [20]. These could be potential ASFV vaccine candidates illegally released into the field [21] or naturally attenuated strains [22]. Further complicating the situation, an attenuated ASFV genotype I strain was reported in 2021 from the Shandong Province in China [23]. The Shandong ASFV strain was a low virulent virus with efficient transmissibility in pigs, causing a chronic disease characterized by necrotic skin lesions and joint swelling. Phylogenetically, the virus showed high similarity to ASFV genotype I viruses isolated in Portugal in 1968 and 1988. In 2023, a possible recombination of these attenuated ASFV genotype I strains with the pandemic ASFV genotype II viruses, generating a highly virulent recombinant strain, was reported in China [24].

The pandemic ASFV p72 genotype II strain is a highly virulent virus that causes peracute or acute disease in infected pigs, characterized by high fever, hemorrhages, loss of appetite, lethargy, and mortality, usually within 7–12 days of exposure [25]. The infected animals shed the virus in body secretions and excretions, and transmit it to contact pigs [26]. In contrast, the ASFV LAV p72 genotype II strains cause delayed disease onset and mild to chronic clinical signs and low mortality [27,28]. The appearance of LAV strains in Asia complicated the ASF clinical presentation, making ASF control and eradication challenging. The subtle clinical signs associated with ASFV LAV strains could easily go unnoticed, leading to widespread disease. Having these emerging ASFV strains reach countries currently free of ASF could have a severe impact on trade. Hence, understanding the types and the extent of emerging LAV and recombinant strains circulating in endemic countries is critical.

This paper describes the identification and characterization of an ASF LAV isolate from a family farm in Northern Vietnam. The isolate was found to be a genetically modified ASFV p72 genotype II strain lacking six genes in the MGF region.

## 2. Materials and Methods

### 2.1. Screening Field Samples for β-GUS Gene

Serum samples collected from family farms in Northern Vietnam that experienced ASF outbreaks between 2019 and 2021 were tested for ASFV genomic material by p72-specific Tignon real-time PCR assay (see below). Serum samples that tested positive for the ASFV genome were subsequently screened for the presence of the β-GUS gene (GUS cassette) by conventional PCR using previously described primers (see below). *E. coli* DH5-alpha genomic DNA was used as the positive control for the GUS conventional PCR. Amplicons were resolved by agarose gel electrophoresis, purified using a QIAquick gel extraction kit (Qiagen, Toronto, ON, Canada), and Sanger sequenced using a BigDye Terminator chemistry version 3.1 (Life Technologies, Burlington, ON, Canada) on an Applied Biosystems 3130xl Genetic Analyzer (Life Technologies).

### 2.2. Virus Isolation and Titration

For virus isolation, GUS PCR positive samples were filtered with Minisart^®^ 0.22 µm Syringe Filters (Sartorius, Goettingen, Germany) and inoculated into primary porcine alveolar macrophage (PAM) and primary porcine leukocyte (PPL) cultures prepared from heparinized whole blood collected from a healthy pig, as described previously [29,30]. The cultures were incubated at 37 °C in a 5% CO_2_ incubator for 7 days, and observed daily under an inverted light microscope for hemadsorption (HAD). On the 7th day, the plates were frozen at −80 °C for at least 4 h, thawed at 4 °C, and the supernatant from each well was subjected to ASFV real-time PCR and GUS conventional PCR, as described below. The supernatants from the samples were passaged once more in PPLs and then subjected to virus titration on primary porcine alveolar macrophage (PAM) cultures. Briefly, 10-fold dilutions of the isolated virus containing the supernatants were inoculated into 90% confluent PAM cells in Minimum Essential Medium Alpha (MEM-α) supplemented with 1% gentamicin, 1% Glutamax, and 2% fetal bovine serum (FBS). Following 3 days of incubation at 37 °C and 5% CO_2_, the plates were fixed and stained with an anti-ASFV polyclonal pig serum (produced in-house) and commercially available horseradish peroxidase (HRP)-conjugated anti-pig goat monoclonal antibody (Jackson Immunoresearch, West Baltimore Pike, West Grove, PA, USA).

### 2.3. Whole-Genome Sequencing, Assembly, and Annotation of the ASFV-GUS-Vietnam Genomes

Total nucleic acids were extracted from the ASFV-GUS-Vietnam isolate, as well as blood and spleen samples collected from pig #1 and pig #14 using a 5× MagMax^TM^ Viral/Pathogen Nucleic Acid Isolation kit (Thermo Fisher Scientific, Waltham, MA, USA) according to manufacturer’s instructions and submitted to NCFAD’s Genomics Unit for high-throughput sequencing, genome assembly, and comparative genome analysis. For Illumina sequencing (n = 1 run), the ASFV genome in the sample was enriched using a custom myBaits^®^ ASF target capture kit (Daicel Arbor Biosciences, Ann Arbor, MI, USA) and the sequence libraries were prepared using a Nextera XT DNA Library Preparation Kit (Illumina, San Diego, CA, USA). Sequencing was performed on an Illumina MiSeq instrument at the NCFAD using a V3 flow cell with a 600-cycle reagent cartridge (Illumina). The same nucleic acid extract was also sequenced five times over 7 months using the GridION sequencer (Oxford Nanopore Technologies, Lexington, MA, USA) to fill in sequencing gaps and GUS transgene and flanking regions, confirm regions with low sequencing depths, and facilitate whole-genome assembly. For Nanopore library preparation, a Rapid PCR barcoding kit SQK-RPB004 (Oxford Nanopore Technologies) was used and the samples were run on R9.4 flow cells (Oxford Nanopore Technologies). A NEB Next Microbiome DNA Enrichment Kit (New England Biolabs, Ipswich, MA, USA) was used on two Nanopore runs to deplete methylated DNA prior to Nanopore library preparation according to the manufacturer’s instructions. An adaptive sampling strategy was used to deplete off-target host reads during the third Nanopore run. The swine reference genome (Sus scrofa; RefSeq: GCF_000003025.6) was used as a reference for host depletion. Nanopore reads were base called using Guppy (two with v6.3.8, one with v6.3.9 and two with v6.4.6) with the super high-accuracy model.

The Illumina reads were first processed with the CFIA-NCFAD/nf-villumina (v2.0.1) [31] and Nextflow pipeline [32] to remove Illumina PhiX Sequencing Control V3 reads with BBDuk [33] and perform quality filtering with fastp [34]. The Nanopore reads were processed with Porechop (v0.2.4) [35] on default settings to remove adapters followed by quality filtering with Chopper (v0.5.0) [36] with the quality score set to 10 and minimum read length set to 50. Trimmed Illumina and Nanopore reads for the ASFV-GUS-Vietnam inoculum sample were mapped to the previously assembled A-MM-A-QI ASFV genome in Geneious Prime (v2023.0.1) [37] with the Geneious assembler on default settings with Low Sensitivity/Fastest sensitivity and 5 iterations to produce a 75% majority consensus sequence with minimum 10x coverage threshold. Nanopore reads were similarly mapped and assembled to produce a second consensus sequence. The consensus sequences were aligned in Geneious Prime (v2023.0.1) with MAFFT (v7.490) [38] on default settings to perform manual alignment-based corrections to produce a final consensus sequence. Consensus sequences for the pig #1 and pig #14 samples were assembled in an identical manner, changing only the assembler used to Minimap2 (v2.24) [39] in Geneious Prime (v2023.0.1).

### 2.4. Animal Experiments

In order to evaluate the virulence of the ASFV-GUS-Vietnam isolate, two pathogenesis experiments were conducted. In both experiments, six- to seven-week-old weaned Large White/Landrace/Duroc cross piglets (each around 15 lbs) obtained from a local supplier in Manitoba were used. The animals were housed in biosafety level 3 (BSL3) animal pens and provided feed (commercial ration) twice a day and water ad libitum. The use of animals was approved by the Animal Care Committee at the Canadian Science Centre for Human and Animal Health (Animal Use Document C-22-001), and all procedures involving animals were in compliance with the Canadian Council for Animal Care guidelines. A clinical scoring system developed in-house (based on published scoring systems for ASF) was used to record the daily clinical findings.

All procedures, except oral fluid collection and measuring rectal temperatures, were performed under general anesthesia through the inhalation of isoflurane 2% delivered in 100% oxygen. Rectal temperatures were taken from individual pigs in the morning after oral fluid collection, under physical restraint or while the pigs were eating. Any pig reaching the humane endpoint was euthanized using a 10 mL sodium pentobarbital (240 mg/mL) intravenous injection. Any animal found dead or euthanized were subjected to a full post-mortem examination and tissue collection for virus detection and histopathology.

In the first experiment, six piglets (#1–6) were housed in a single pen and after 14 days of acclimatization, were inoculated oro-nasally (ON) with cell culture-amplified ASFV-GUS-Vietnam virus (1 × 10^5^ TCID_50_/mL), 1 mL orally and 0.5 mL in each nostril for a total dose of 2 × 10^5^ TCID_50_/pig. After inoculation, the pigs were observed at least twice per day (morning and afternoon) and with greater frequency as the animals developed clinical signs. Every day starting at 0 dpi, before feeding the pigs, oral fluids were collected from the pen utilizing cotton ropes from the TEGO^®^ Swine Oral Fluid kit (ITL Biomedical, Reston, VA, USA) as described previously [40]. Whole blood (EDTA), oropharyngeal swabs (OPSWs), buccal swabs (BSWs), rectal swabs (RSWs), and nasal swabs (NSWs) were collected from individual animals every third day. For swab collection, polyester-tipped polystyrene applicators (Puritan^®^ Medical Products, Guilford, New York, NY, USA) in 1 mL of sterile Dulbecco’s Phosphate-Buffered Saline (DPBS) were used (Corning Mediatech Inc., Manassas, VA, USA). At 48 days post inoculation (48 dpi) with ASFV-GUS-Vietnam, the surviving pigs were challenged intramuscularly (IM) with 1 mL (0.5 mL into each thigh) of ASFV Georgia 2007/1 (1 × 10^3^ TCID_50_/mL) for a total dose of 1 × 10^3^ TCID_50_/pig. As controls, two naïve, age- and breed-matched pigs (#7 and 8) housed in a separate pen were inoculated IM with the same dose of ASFV Georgia 2007/1. After the challenge, the animals were monitored for a minimum of two times per day (morning and afternoon), and with greater frequency as the animals developed clinical signs. On day 69 post primary inoculation with ASFV-GUS-Vietnam, i.e., 21 days post challenge (dpc) with ASFV Georgia 2007/1, all surviving pigs were euthanized using a 10 mL sodium pentobarbital (240 mg/mL) intravenous injection, and subjected to full post-mortem examinations with various tissue samples collected.

In the second experiment, a total of six piglets (same age, breed, and supplier) were housed in a single pen. After 14 days of acclimatization, four (#9-12) out of six piglets were inoculated with 1 mL of ASFV-GUS-Vietnam (2 × 10^3^ TCID_50_/mL) IM into the left hamstring for a total dose of 2 × 10^3^ TCID_50_/pig. The remaining two piglets (#13 and 14) were inoculated with sterile DPBS in the same manner (1 mL per pig IM). After the inoculation, the pigs were observed a minimum of twice per day (morning and afternoon). Similar to the first experiment, starting on 0 dpi, oral fluids were collected from the pen every day, and whole blood (EDTA), OPSWs, BSWs, RSWs, and NSWs were collected from individual animals every third day as described above. On 44 dpi, each surviving pig was challenged with 2 mL of ASFV Georgia 2007/1 (1 × 10^5^ TCID_50_/mL) ON (1.0 mL orally and 0.5 mL in each nostril) for a total dose of 2 × 10^5^ TCID_50_/pig. After the challenge, as described previously, the animals were monitored at least twice a day and with greater frequency as the animals developed clinical signs. On day 69 post primary inoculation (21 dpc), all surviving pigs were euthanized using a 10 mL sodium pentobarbital (240 mg/mL) intravenous injection, and subjected to full post-mortem examinations.

### 2.5. DNA Extraction, Conventional PCR, and Real-Time PCR Assays

Total nucleic acids were extracted from whole blood, swabs, oral fluids, and 10% (*w*/*v*) tissue homogenates using MagMax™ Pathogen RNA/DNA Kit (Thermofisher Scientific, Waltham, MA, USA) according to the protocol provided by the manufacturer on a MagMax™ Express-96 Deep Well Magnetic Particle Processor(Thermofisher Scientific, Waltham, MA, USA). A sample volume of 55 µL (blood and tissue homogenates) or 300 µL (low cell count samples—swabs and oral fluids) was used according to the manufacturer’s instructions.

To detect the GUS gene inserted into the ASFV genome, a conventional GUS PCR assay that has been described previously was used [41]. The cycling conditions were as follows: 10 min initial denaturation at 95 °C followed by 40 cycles of 15 s denaturation at 95 °C, 15 s annealing at 55 °C, 30 s extension at 72 °C, and a 5 min final extension at 72 °C. A conventional PCR targeting the MGF 505-2R gene was designed in-house (FP 5′-GCAGAGGTATGATGTCCTTA-3′ and RP 5′-AGATACTTGTTCAACAGGAA-3′). The cycling conditions for MGF 505-2R PCR were as follows: 3 min initial denaturation at 95 °C followed by 40 cycles of 30 s denaturation at 95 °C, 30 s annealing at 56 °C, 1 min extension at 72 °C, and a 10 min final extension at 72 °C. Both conventional PCR assays used DreamTaq Green PCR Master Mix (ThermoFisher Scientific).

ASFV genomic DNA was detected using three different quantitative real-time PCR assays. The Tignon PCR assay [42], which targets a highly conserved region of the p72 gene, was used to test all of the samples. The modified Zsak PCR assay [43] targeting the ASFV p72 gene was also used to test the oral fluid samples. The Ct values were converted to viral genomic log_10_ copy numbers based on a standard curve. For differentiation of ASFV-GUS-Vietnam from ASFV Georgia 2007/1, a real-time PCR assay targeting the MGF 360-14L gene (FP 5′-AGAAGACGGGGTTCGGATACAG-3′, RP 5′-GCAAATCCTGAATATGGGCTTATACG-3′, and Probe 5′-CTCCCAGTTCCGCACACAGCCGC-3′) was used [44]. Using an ASFV Georgia 2007/1 dilution series, the sensitivity of the MGF 360-14L real-time PCR assay was compared to that of the Tignon assay (Appendix A). ß−actin was used as the internal extraction and amplification control [45]. All real-time PCR reactions were prepared using Taqman^®^ Fast 1-Step Master Mix (Thermofisher Scientific) and run on a Bio-Rad CFX96 instrument (Bio-Rad Laboratories Canada Ltd., Mississauga, ON, Canada) using the fast amplification protocol (50 °C for 5 min; 95 °C for 20 s, 95 °C for 3 s and 60 °C for 30 s).

In order to detect ongoing endemic viral infections, a selected number of serum and tissue samples from the pigs were tested using published real-time PCR assays specific for porcine teschovirus, sapelovirus [46], and circovirus 2 [47].

### 2.6. Enzyme-Linked Immunosorbent Assay (ELISA)

The INgezim PPA COMPAC blocking ELISA (Gold Standard Diagnostics, Madrid, Spain) was used for testing the sera for anti-ASFV antibodies according to the manufacturer’s instructions. Briefly, the pre-coated PPA plate was allowed to equilibrate to room temperature for at least 30 min. Then, 100 µL of a 1/2 dilution (made with the included diluent) of the controls or test sera was applied to the plate in duplicate. The plate was sealed and incubated for one hour at 37 °C. The samples were removed, and the plate was washed four times using the included washing solution. Peroxidase-labelled monoclonal antibody (mAb) specific to ASFV p72 was diluted to 1/100 and applied at a volume of 100 µL per well. The plate was incubated for 30 min at 37 °C. The plate was then washed five times. The TMB substrate was applied at a volume of 100 µL per well and allowed to develop for 15 min in the dark. The stop solution (100 µL per well) was applied at the end of the incubation time, and the plate was read at 450 nm within 5 min of completion. The blocking percentage was calculated as follows: Blocking % (× %) of a sample = ((NC − Sample OD)/(NC − PC)) × 100. Result interpretation: IP ≥ 50%, positive; IP < 40%, negative; 50% > IP ≥ 40%, suspicious.

## 3. Results

### 3.1. Detection of β-Glucuronidases Gene in ASFV-Positive Serum Samples Collected from Vietnam

One of the two serum samples (collected on 26 September 2021 from Hung Yen Province) that tested positive for both ASFV p72-specific real-time PCR and conventional PCR specific for the GUS cassette resulted in HAD when inoculated into PPLs and PAMs (Figure 1A). The isolate ASFV-GUS-Vietnam was re-tested for the presence of the GUS cassette by conventional PCR (Figure 1B). The conventional PCR resulted in a 471 bp product with ASFV-GUS-Vietnam and *E. coli* DNA as expected [41]. The amplicons were Sanger sequenced to confirm their identity. The conventional GUS PCR also resulted in a smaller amplicon when nucleic acid extracted from ASFV Georgia 2007/1 propagated in PPLs was used, and Sanger sequencing revealed that the amplicon is Sus scrofa genomic DNA from chromosome 17 (GenBank # CP071568), confirming a non-specific amplification.

### 3.2. Clinical Signs in Pigs Inoculated ON with ASFV-GUS-Vietnam and Challenged IM with ASFV Georgia 2007/1

Following the inoculation with ASFV-GUS-Vietnam, pigs #1–6 stayed active, had a normal appetite, and appeared normal until 7 dpi. Starting on 7 dpi, the rectal temperatures of all pigs, except #5, were slightly elevated. On 9 dpi, pig #1 was depressed, shivering, lying down, and did not chew the rope. When encouraged to stand, he walked a short distance with a stiff gait. Pig #1 had blood on both the pharyngeal and nasal swabs, and was bleeding from both nostrils for a short period of time after swabbing. Pig #6 had a small amount of blood on the rectal swab. On 10 dpi, most of the feed was left uneaten. Pig #1 was shivering, had piloerection, and hyperemic ears, tail, and rump. It had conjunctivitis in left eye, did not chew the rope, and was mildly ataxic. The pig ate fresh food when offered but not as eagerly as other pigs. The purplish discoloration of the ears on this pig remained the same.

When observed through the CCTV camera, pig #1 was seen getting up and walking around, eating a bit of food, and nosing the other pigs occasionally. On 11 dpi, less feces was observed in the pen and some food was leftover in the feed bins. Pig #5 was slow to get up. Pig #1 was very reluctant to get up, shivering, ataxic, and showed no interest in the rope and little interest in food. It also had purplish discoloration of the ear, tail, and rump. When encouraged, it got up, ate a small amount of feed, and drank a bit of water. On 12 dpi, pigs #4, 5, and 6 were bright, alert, and responsive and pigs # 2 and 3 were quiet but active and responsive. Pig #1 was severely depressed, unable to stand, dyspneic, and its rectal temperature decreased below normal (hypothermia). Cyanosis in the ear tips, and echymotic hemorrhages in the tail, rump, and abdomen were observed with blood present on the nasal and rectal swabs. Pig #1 was euthanized later in the day, as it reached the humane endpoint.

On 13 dpi, some feed was left in the pen overnight and mashed into the floor. A reduced amount of feces was observed and pig #2 was hunched, displaying piloerection and a slightly stiff gait. Pig #6 appeared slightly emaciated. All pigs went to the rope and chewed, ate eagerly when offered fresh feed, and remained standing while cleaning the cubicle. Through the CCTV, pig #6 appeared to startle, spin rapidly, and fall over. It took some time for the pig to stand, settle, and return to feed. This was observed for a second time approximately 1 min later.

On 14 dpi, pig #2 was slightly hunched but active. All pigs ate eagerly when offered fresh feed. The rectal temperatures of all five pigs were in the normal range. By the end of the day, all pigs were active and appeared normal. Starting at 15 dpi, the pigs appeared normal, stayed active, and had normal rectal temperatures.

At 48 dpi, each pig was challenged IM with ASFV Georgia 2007/1 (1 × 10^3^ TCID_50_/pig). After the challenge, the pigs remained active, alert, and had normal rectal temperatures. All five pigs were euthanized on 69 dpi (21 dpc), concluding the study.

Two naïve pigs, (#7 and 8), age- and breed-matched, were housed in a separate pen as control pigs. When challenged with the same dose of ASFV Georgia 2007/1 IM, they developed a high fever by 4 dpc. Feed was left in the pen overnight and dry feces was observed in the pen. Pig #7 had to be encouraged to stand but it laid down again shortly after. Both pigs were not interested in the rope but ate when offered fresh feed. The pigs were observed through a CCTV camera several times throughout the day and they were inactive, shivering occasionally, off feed and mostly lying down. On 5 dpc, both pigs remained obtunded and had a high fever. On 6 dpc, they reached the humane endpoint and therefore, were euthanized.

The cumulative clinical score (CCS), which indicates the seriousness of the illness, was calculated for each animal every day (Appendix A). In line with the transient mild clinical signs observed, the CCSs of pigs #2–6 showed a transient increase (<5) between 7 and 14 dpi. In contrast, the CCS of pig #1 continued to increase and reached its maximum by 12 dpi. After the challenge, the CCSs of pigs #2–6 showed no increase, whereas the CCSs of control pigs #7 and 8 started to rise on 2 dpc, and reached their maximum values by 6 dpc.

### 3.3. Viremia and ASFV Shedding in Pigs Inoculated ON with ASFV-GUS-Vietnam and Challenged IM with ASFV Georgia 2007/1

Following the inoculation (ON) of ASFV-GUS-Vietnam (2 × 10^5^ TCID_50_/pig), five of the six pigs developed viremia, coinciding with fever (Figure 2). The viremia was first detected on 6 dpi in three pigs (#1, 2 and 4) and they reached peak viremia on 9 dpi (Figure 3A). Pig #1, which was euthanized on 12 dpi, developed the highest viremia. Pig #5 did not develop viremia or a fever. Pigs #4 and 6 developed viremia on 9 dpi. Although the fever subsided in pigs #2–4 and 6 (Figure 2), they showed intermittent viremia until the end of the study (Figure 3A). ASFV DNA was detected in the blood of pig #5 twice after the challenge (IM) with ASFV Georgia 2007/1 (Figure 3A).

In the swabs, the earliest detection of ASFV genomic material was on 6 dpi (pig #3, NSW; Figure 3D). By 9 dpi, ASFV genomic material was detected in OPSWs (pigs #1, 2 and 6; Figure 3B), BSWs (pigs #4 and 6; Figure 3C), NSWs (pigs #1 and 3; Figure 3D), and an RSW (pig #1; Figure 3E). Thereafter, the ASFV genome was detected in swab samples intermittently until the end of the experiment. Two swab samples (NSW and BSW) collected at 27 dpi from pig #5, which did not develop viremia before the challenge, also displayed ASFV genomic material (Figure 3C,D).

The two control pigs (#7 and 8) rapidly developed viremia after the challenge, and all swab samples collected tested positive for ASFV genomic material (Figure 3B–E).

The aggregate oral fluid samples were collected only from the pen that housed pigs #1–6, and not the control pigs. ASFV genomic material was detected in the oral fluid samples starting on 7 dpi (Ct 34.4 in modified Zsak assay), five days before pig #1 reached the humane endpoint (Figure 3F). The highest amount of the ASFV genome in oral fluids was observed on 13 dpi, one day after pig #1 was euthanized. Despite the absence of clinically sick pigs in the pen, the ASFV genome was intermittently detected in the oral fluids and it appeared to coincide with viremia in the pigs ON inoculated with ASFV-GUS-Vietnam. The modified Zsak assay provided lower Ct values (higher ASFV genome copies) than the Tignon assay in the oral fluid samples.

ASFV genomic material-positive (based on the Tignon assay) whole blood and OPSW samples (except the OPSW collected on 7 dpc from pig #4) tested negative on the MGF 360-14L real-time PCR assay (Appendix A).

### 3.4. Anti-ASFV Antibody Detection in Pigs Inoculated ON with ASFV-GUS-Vietnam and Challenged IM with ASFV Georgia 2007/1

All pigs that survived the ON infection with ASFV-GUS-Vietnam tested positive for antibodies to ASFV (Figure 4). All surviving pigs, except pig #5, tested positive by ELISA by 11 dpi, and pig #5 was positive at 20 dpi. The sera from pigs #1, 7, and 8 did not test positive by ELISA.

### 3.5. Clinical Signs in Pigs Inoculated IM with ASFV-GUS-Vietnam and Challenged ON with ASFV Georgia 2007/1

In the second experiment, after the acclimatization period, four out of six piglets (pigs #9–12) were inoculated IM with ASFV-GUS-Vietnam at a dose of 2 × 10^3^ TCID_50_/pig. The remaining two pigs (#13–14) received sterile DPBS IM. All four pigs that received ASFV-GUS-Vietnam IM developed a fever within 7 dpi (Figure 5). On 5 dpi, pigs #10, 11, and 12 developed a mild fever (rectal temperature above 40 °C). On 6 dpi, pig #11 had a mild fever, but not the others. On 7 dpi, pigs #9 and 11 had rectal temperatures of 40.4 °C and 41.3 °C, respectively. On 8 dpi, pigs #11 and 12 had a mild fever (both at 40.6 °C). On 9 dpi, pigs #10 and #12 had rectal temperatures of 40.5 °C and 41.1 °C, respectively. Despite having a fever, all pigs remained active and appeared normal. On 10 dpi, dry pelleted feces (~75%) were observed in the pen and some fecal pellets were stained with blood. Pig #9 was not interested in chewing the rope, appeared thin, but ate readily when fresh feed was provided. Pig #10 was less active and was observed lying down frequently. On 11 dpi, all four animals developed a fever, and pig #9 was less active, not very interested in chewing the rope, appeared thin, and was slightly pale. The feces within the pen remained dry and pelleted.

On 12 dpi, all pigs inoculated (IM) with ASFV-GUS-Vietnam had a mild fever (40.5 °C to 40.9 °C), and pig #9 remained less active, not very interested in chewing the rope, and appeared thin and slightly pale. Pig #10 was less active walked with a hunched back and slightly stiff gait. On 13 dpi, pigs #10 and 12 had increased rectal temperatures. Pig #9 continued to appear thin and was noticeably smaller than the other pigs; however, it readily ate when feed was offered. Pig #10 continued to be hunched back and appeared to have difficulty getting up and preferred to be seated (dog sitting posture). On the morning of day 14, several spots of white vomit were noticed on the floor. Pig #9 appeared thin but readily ate when fresh feed was offered. Pig #10 remained seated but a slight improvement was noticed. Pig #11 was less active and pig #12 was observed heaving slightly, then chewing and swallowing afterwards and was therefore suspected to have vomited earlier.

By 15 dpi, pig #9 appeared to gain some weight back, and pig #10 also appeared to continue to improve. However, pig #11 remained obtunded. Blood was observed on rectal swabs from pigs #9 and 13. The feces of pigs #13 and 14 were dry and pelleted. None of the pigs had increased rectal temperatures and no vomitus was observed. On 16 dpi, several spots of white vomitus were observed again on the floor. Pig #9 was observed vomiting once. Pig #14 developed a mild fever (40.8 °C) and was less active and hyperemic.

On 17 dpi, pig #10 appeared to continue to improve. Both contact pigs (#13 and 14) had a high fever of 41.1 °C and 41.6 °C, respectively, and remained hyperemic. The fever lasted for the next two days. By 20 dpi, pig #13 appeared to recover and had a rectal temperature of 40.3 °C, but pig #14 remained feverish (41.3 °C) and hyperemic. On 21 dpi, pig #14 remained less active and feverish (41.3 °C); it was occasionally shivering but ate normally. On 22 dpi, all pigs except #14 were active. Pig #14 was less active and ate normally but had slight drop in rectal temperature (40.6 °C). On 23 dpi, pig #14 was found dead. A full necropsy was performed and various tissue samples were collected.

The remaining pigs stayed healthy, bright, and alert and on 44 dpi, they were challenged with ASF Georgia 2007/1 (2 × 10^5^ TCID_50_/pig) ON in 2 mL (1 mL orally and 0.5 mL in each nostril). Following the challenge, all pigs (#9–13) remained healthy, and appeared normal, bright, and alert until the end of the experiment. On 69 dpi (25 dpc), all pigs were humanely euthanized and the study was concluded.

Two age- and breed-matched control pigs (#15 and 16) that were challenged via the ON route in the same manner with the same dose as pigs #9–13 above, developed a fever on 5 dpc. On 6 dpc, dry feces and leftover feed in the pen were observed, and the rectal temperatures of both pigs increased to 41.6 °C. By the end of the day, fresh feces covered with blood clots were observed in the pen. On 7 dpc, both pigs were euthanized as they reached the humane endpoint.

In line with the mild clinical signs observed after the ASFV-GUS-Vietnam IM inoculation, the individual CCSs of pigs #9–12 transiently increased (Appendix A). In contrast, the CCS of contact pig #14 continued to increase and reached its maximum by 23 dpi. After the challenge with ASFV Georgia 2007/1 ON, the CCSs of the control pigs increased and reached their maximum values by 7 dpc.

### 3.6. Viremia and ASFV Shedding in Pigs Inoculated IM with ASFV-GUS-Vietnam and Challenged ON with ASFV Georgia 2007/1

Following the IM inoculation of ASFV-GUS-Vietnam, all four pigs developed viremia which coincided with fever. Viremia was first detected in the pigs on 6 dpi (Figure 6A). The genome copy numbers of ASFV in the blood of these pigs were higher than those that were inoculated ON with ASFV-GUS-Vietnam in the first experiment (Figure 6A). The highest viremia in the group was detected in pig #9 on 6 dpi, but the viremia decreased rapidly over time. Similar to the observations in the first experiment, the pigs inoculated IM with ASFV-GUS-Vietnam showed intermittent viremia until the end of the study, despite the absence of clinical signs.

ASFV genome was detected starting from 6 dpi in the swab samples from all pigs that received ASFV-GUS-Vietnam IM. The amount of ASFV genomic material in the swab samples decreased over time. However, intermittent detection in all swab types was observed until the end of the experiment. The OPSWs, NSWs, and RSWs tested positive for the ASFV genome starting on 6 dpi (Figure 6B,D,E), and BSWs were positive starting from 9 dpi (Figure 6C). The OPSWs and RSWs from pig #9 contained the highest ASFV genome copy numbers among those that received ASFV-GUS-Vietnam IM. Overall, the swabs from pigs that received ASFV-GUS-Vietnam IM (2 × 10^3^ TCID_50_/pig) had higher ASFV genome copy numbers than those from pigs that received the same virus via the ON (2 × 10^5^TCID_50_/pig) route.

Among the two contact pigs, viremia was first detected in pig #14 on 15 dpi, a day before it developed a fever (Figure 6A). Viremia was first detected in the second contact pig (#13) on 17 dpi, the same day it developed a fever. Pig #14, which died on 23 dpi, had the highest ASFV genome copy number in the blood among those that were exposed to ASFV-GUS-Vietnam. The viral load in the blood of pig #13 started to decrease starting from 27 dpi. The swab samples from pig #14 had more copies of the ASFV genome than the swabs collected from pig #13 that survived. All four types of swab samples from pig #13 intermittently indicated the presence of ASFV genomic material until the end of the experiment. Among the swab types, the OPSWs had the highest ASFV genome copy numbers from pig #13 (4.23 on 43 dpi).

All four ASFV-GUS-Vietnam IM inoculated pigs and one of the contact pigs that survived continued to show intermittent viremia and all pigs (except pig #9) shed virus at least once. The whole blood and OPSW samples that contained ASFV genomic material (based on the Tignon assay) tested negative by the MGF 360-14L real-time PCR assay (Appendix A).

Oral fluids collected from the pen that housed pigs #9–14 showed the presence of the ASFV genome starting on 7 dpi (Ct 32.44 in modified Zsak assay), 16 days before the contact pig (#14) died (Figure 6F).

ASFV genome copy number in oral fluids peaked twice before pig #14 died, and they coincided with a fever and two peaks of viremia, one due to infection of pigs #9 to 12 (IM inoculation), and the second due to the infection of pigs #13 and 14 by contact transmission, most likely through the ON route with ASFV-GUS-Vietnam. Another increase in detection of ASFV genomic material in oral fluids was around the period of the challenge infection. As seen in experiment 1, the modified Zsak assay resulted in lower Ct values than the Tignon assay. ASFV genomic material continued to be detected in oral fluid samples for about a week after the challenge.

The control pigs (#15 and 16) challenged with ASFV Georgia 2007/1 developed viremia by 4 dpc and reached the highest titer by 7 dpc before they were euthanized (Figure 6A). All four swab types from the challenged control pigs tested positive for ASFV genome by the Tignon assay, and the copy numbers of the final swabs were higher than those detected in pigs that received ASFV-GUS-Vietnam IM.

### 3.7. Anti-ASFV Antibody Detection in Pigs Inoculated IM with ASFV-GUS-Vietnam and Challenged ON with ASFV Georgia 2007/1

All pigs (#9–12) that received ASFV-GUS-Vietnam via the IM route developed antibodies to ASFV (Figure 7). Suspicious levels of anti-ASFV antibodies were detected in pigs #9, 11, and 12 on 9 dpi. Anti-ASFV antibodies were detected in pig #10′s serum starting on 13 dpi. The contact pig that died of ASF did not develop detectable levels of antibodies to ASFV, similar to the two control pigs (#15 and #16) that received ASFV Georgia 2007/1 via the ON route. After the challenge, the anti-ASFV antibody titers in all remaining pigs increased.

### 3.8. Post-Mortem Findings of Pigs That Succumbed to ASFV-GUS-Vietnam Infection

Pig #1, inoculated ON with ASFV-GUS-Vietnam (total dose of 2 × 10^5^ TCID_50_), was euthanized on 12 dpi, as it reached the humane endpoint. External examination revealed cyanotic ear tips (Figure 8A), ecchymotic skin hemorrhages on the ventral abdomen (Figure 8B) and flanks, and small bruises on the neck and tail. A slight bloody nasal discharge from both nostrils was also observed. When the carcass was opened, watery/thin blood was observed. The spleen was enlarged (Figure 8D) and petechial hemorrhages were observed on the serosal surface of the stomach, pancreas, and intestines. The gastro-hepatic lymph nodes were enlarged and hemorrhagic (Figure 8H). The lungs were heavily congested (Figure 8F) and a serosanguineous pericardial effusion (Figure 8E) was observed. Epicardial hemorrhages (Figure 8F) were observed and in some areas, the hemorrhages extended across the myocardium. The superficial inguinal lymph nodes were edematous, and a few focal hemorrhages were observed.

Pig #14 was one of the two contact pigs in the second experiment. It was found dead on 23 dpi. During the external examination, ecchymotic hemorrhages on the ventral abdomen and fresh blood in the rectum were observed. When the abdominal cavity was opened, a large volume of serosanguineous peritoneal fluid was evident (Figure 8C). The renal lymph nodes were hemorrhagic, and hemorrhages were observed in the renal medulla. Petechial hemorrhages on the renal cortex (Figure 8J) and the gall bladder were also observed (Figure 8K). Similar to pig #1, an enlarged spleen, enlarged and hemorrhagic gastro-hepatic lymph nodes, serosanguineous blood in the pericardium, and massive epicardial hemorrhages were observed. The lungs were slightly congested and petechial hemorrhages were observed across the left and right lung lobes. Petechial and echymotic hemorrhages were also observed in the diaphragm and the submandibular lymph nodes were enlarged and severely congested (Figure 8G).

The bone marrow from pigs #1 and 14 was tested by PCR for the presence of GUS and the absence of the MGF 505-2R genes (Figure 9). The GUS amplicon was detected in the bone marrow samples from pigs #1 and #14. No MGF 505-2R amplicon was detected in the ASFV-GUS-Vietnam inoculum or in the bone marrow samples from pigs #1 and #14, suggesting that wild-type (ASFV genotype II epidemic strain circulating in Vietnam) ASFV was not responsible for the fatal outcome observed in these two pigs. The same samples also tested negative for MGF 360-14L by real-time PCR (Table 1).

A number of other lymphoid tissues from the pigs were tested by both Tignon and MGF 360-14L real-time PCR and none of them were positive for MGF 360-14L (Appendix A).

### 3.9. Whole-Genome Sequencing, Assembly, and Annotation of the ASFV-GUS-Vietnam Genomes

The nucleic acids extracted from the ASFV-GUS-Vietnam inoculum and blood and spleen samples collected from pigs #1 (ASFV-GUS-Vietnam-1) and 14 (ASFV-GUS-Vietnam-14) were subjected to whole-genome sequencing using both Nanopore technology and Illumina MiSeq. All three genomes were fully sequenced, annotated, and submitted to GenBank. The sequences of the three viruses, ASFV-GUS-Vietnam (GenBank: PP213439), ASFV-GUS-Vietnam-1 (GenBank: PP213440), and ASFV-GUS-Vietnam-14 (GenBank: PP213441), were compared to the ASFV Georgia 2007/1 sequence available in GenBank (FR682468) (Table 2). The whole-genome sequencing data confirmed a GUS cassette insertion and deletion of six genes in the MGF region (MGF505-1R, MGF360-12L, MGF360-13L, MGF360-14L, MGF505-2R, and MGF505-3R) in the genomes of all three viruses. The deletion was identical to that described for the live-attenuated recombinant strain described by O’Donnell et al., 2015 [48]. Compared to ASFV Georgia 2007/1, the three ASFV-GUS-Vietnam viruses (inoculum and from pigs #1 and #14) had a number of mutations throughout the genome; however, most of them were silent or in the non-coding regions. The mutation observed at nucleotide position 7013 extended the ORF of MGF-110-1L in all three GUS viruses. The deletion of the three nucleotides resulted in the loss of an amino acid in the MGF-110 10L and MGF110 14L proteins in all three GUS viruses. The whole-genome sequences of ASFV-GUS-Vietnam, ASFV-GUS-Vietnam-1, and ASFV-GUS-Vietnam-14 were almost identical.

### 3.10. Detection of Porcine Teschovirus, Sapelovirus, and Circovirus 2 in Serum and Tissue Samples

The serum samples and tonsil samples from pigs #1–16 tested negative for porcine sapelovirus. The tonsil samples from pigs #1–6 tested positive for porcine circovirus 2 (PCV2) (Ct values ranging from 11.2 to 19.6), and the serum samples from pigs #1 (Ct = 25.3), 2 (Ct = 27.6), 3 (Ct = 30.24), and 6 (Ct = 25.9) were positive for PCV2. The tonsil samples from pigs #1 (Ct = 28.27), 2 (Ct = 27.8), 3 (Ct = 32.5), 5 (Ct = 29.7), 6 (32.5), 8 (Ct = 33.5), 11 (Ct = 30.8) and 12 (Ct = 31.1) tested positive for teschovirus. Except for the serum sample from pig #5 (Ct = 34.7), all the other serum samples were negative for teschovirus.

## 4. Discussion

Since the introduction of ASFV into Asia, the emergence of naturally attenuated or illegally introduced attenuated ASFV strains has been a major concern. These concerns later became a reality when China reported the identification of live-attenuated ASFV strains circulating in pig farms. Vietnam and China share a 1281 km (796 mi) border, and many illegal pork products and live animals traverse the border. In order to identify a potential introduction of genetically modified strains to Vietnam, we screened clinical samples obtained from pig farms in the northern provinces of Vietnam. *E. coli* ß-glucuronidase encoded by the GusA gene (GUS) is a widely used marker/reporter for generating recombinant viruses when traditional homologous recombination techniques are used [41,48,49]. We identified two serum samples (out of 414 serum samples tested) that were positive for both the ASFV genome and GUS gene, which were obtained from a family farm in Northern Vietnam, and successfully obtained a virus isolate (ASFV-GUS-Vietnam) from one of these samples.

To characterize ASFV-GUS-Vietnam, we conducted two animal experiments. The first animal experiment was conducted before the whole-genome sequence of the isolate was available. In the first experiment, the ASFV-GUS-Vietnam isolate was inoculated oro-nasally into six weaned pigs at a dose of 2 × 10^5^ TCID_50_/pig, as per the NCFAD pathogenesis study protocol. All six pigs were playful and bright until 7 dpi, suggesting that the ASFV-GUS-Vietnam isolate is an attenuated strain. However, starting at 7 dpi, the pigs started to show ASF-related clinical signs; as the clinical signs got worse, viremia was detected in the inoculated pigs. Pig #1 developed severe clinical signs and was euthanized on 12 dpi. The post-mortem findings of pig #1 aligned with an acute ASF infection, and the whole blood and tissue samples collected from the pig tested positive for GUS. The remaining five pigs recovered by 15 dpi, but some continued to show low levels of intermittent viremia and shedding. When they were challenged IM with the highly virulent ASFV Georgia 2007/1 strain, none of the pigs developed clinical signs of ASF. However, some animals continued to shed virus intermittently. In contrast, when the two age- and breed-matched control pigs were challenged IM with the highly virulent ASFV Georgia 2007/1 strain, they developed severe clinical signs and were euthanized by 6 dpc.

By the end of this study, we were able to complete the whole-genome sequencing of the ASFV-GUS-Vietnam isolate, and it was evident that the virus is a genetically modified ASFV p72 genotype II strain lacking six genes (MGF505-1R, MGF360-12L, MGF360-13L, MGF360-14L, MGF505-2R, and MGF505-3R) in the MGF region, similar to what was described previously for the ASFV-G-ΔMGF virus [41]. The whole-genome sequencing data did not show any evidence of wild-type (ASFV genotype II epidemic strain circulating in Vietnam) ASFV contamination in the inoculum. The results were further confirmed by conventional and real-time PCR targeting MGF 505-2R and MGF 360-14L, two of the missing genes in the ASFV-GUS-Vietnam inoculum. Furthermore, using MGF 360-14L real-time PCR, the absence of a wild-type ASFV strain in the blood samples collected before and after the challenge and lymphoid tissues collected at the end of the study was confirmed (Appendix A). When the swab samples were tested for MGF 360-14L, one sample (OPSW collected on 7 dpc from pig #4) tested positive, and this could be from potential environmental contamination at the time of the IM challenge.

The clinical signs observed in this study were slightly different from those reported for ASFV-G-ΔMGF previously. In 2015, O’Donnell et al. observed no clinical signs when ASFV-G-ΔMGF (1 × 10^2^ or 1 × 10^4^ 50% hemadsorbing doses—HAD_50_) was inoculated IM into 80-90-pound (10–12-week-old) domestic pigs [48]. However, despite the absence of clinical signs, most of the pigs developed low levels of viremia. In a recent study when ten 6–8-week-old pigs were infected IM with ASFV-G-ΔMGF (1 × 10^3^ or 1 × 10^4^ HAD_50_), none of the pigs developed ASF clinical signs other than fever in one of the animals on 12 dpi [50]. In both these experiments, the pigs were inoculated IM and therefore the IM route of inoculation was selected for our second experiment.

In the second experiment, four domestic pigs at seven weeks of age were inoculated IM with ASFV-GUS-Vietnam at a dose of 2 × 10^3^ TCID_50_/pig, i.e., 100 times lower than that dose used for the oro-nasal inoculation. Since low levels of ASFV-GUS-Vietnam shedding was observed in the first experiment, we included two contact pigs in the second study to assess possible horizontal transmission. All four IM inoculated pigs developed a fever within 7 dpi, became lethargic, and lost their appetite but slowly recovered. Compared to inoculation via the oro-nasal route, the viremia and shedding observed following the intramuscular route was higher. The two contact pigs developed a fever and viremia, confirming horizontal transmission, and one of them (pig #14) succumbed to ASF. The clinical signs and post-mortem findings of pig #14 were similar to those of pig #1 in the first experiment.

When challenged, all four IM inoculated and contact pigs that survived did not develop any clinical signs and were fully protected from the lethal challenge. These animals, however, continued to shed low levels of ASFV-GUS-Vietnam intermittently (Appendix A), and at the end of the study, the ASFV-GUS-Vietnam genome was detected in the lymphoid organs of some pigs (Appendix A). In these animals, there was no evidence of the challenge virus in the blood (Appendix A) or swab samples collected during the study (Appendix A), or in tissue samples (Appendix A) collected at the end of the study.

The increased clinical signs and the death of two pigs observed in the experiments with ASFV-GUS-Vietnam were not reported for ASFV-G-ΔMGF. ASFV-G-ΔMGF is derived from an ASFV isolate (ASFV-G) obtained from Georgia during the 2007 outbreak. The origin of ASFV-GUS-Vietnam is unknown. It was isolated from a serum sample collected in 2021, before the ASFV-G-ΔMGF-based vaccine was licensed in Vietnam. When the genome of the ASFV-GUS-Vietnam isolate was compared to whole-genome sequence of ASFV Georgia 2007/1 that is available in GenBank, several mutations were observed outside the MGF region. These mutations could have contributed to the increased virulence of ASFV-GUS-Vietnam observed in this study. This argument is supported by the findings from a recent reversion to virulence in an in vivo study conducted using the ASFV-G-ΔMGF strain that was grown in a proprietary cell line [51]. In that study, the cell line-adapted ASFV-G-ΔMGF was serially passaged in pigs five times using a strict procedure looking at the highest titers found in target organs. Although the virus did not revert to virulence, a variant associated with transient fevers and increased replication and shedding was observed, along with genomic changes.

One can also argue that the death of pigs #1 and #14 could be due to the reversion to virulence of the ASFV-GUS-Vietnam strain. The whole-genome sequences of ASFV-GUS-Vietnam, ASFV-GUS-Vietnam-1, and ASFV-GUS-Vietnam-14 were almost identical (99.99%), and no large deletions, insertions, duplications, or rearrangements were observed; therefore, it is highly unlikely that ASFV-GUS-Vietnam reverted to a virulent phenotype in pigs #1 and #14. The results from the MGF 505-2R conventional and MGF 360-14L real-time PCRs further confirmed no involvement of wild-type virus in the fatal disease outcome of the two pigs.

The death of the two pigs infected with ASFV-GUS-Vietnam could also be due to host factors such as the age, genetic makeup (breed), and underlying health conditions of the pigs used in the experiments. The pigs used in the current study were purchased from a porcine reproductive and respiratory syndrome virus (PRRSV)-free high-health farm in Manitoba, Canada. They appeared healthy when purchased and stayed healthy until the ASF clinical signs appeared after the inoculation of ASF-GUS-Vietnam. At the post-mortem assessment, no obvious gross lesions indicative of co-infections was observed; however, some pigs tested positive for teschovirus. All tonsil samples and some serum samples from six pigs ON inoculated with ASFV-GUS-Vietnam also tested positive for PCV2 by PCR. PCV2 leads to lymphoid depletion and immunosuppression and this could have contributed to the fatal ASFV-GUS-Vietnam infection in pig #1. None of the pigs in experiment 2 tested positive for PCV-2 and therefore, it was unlikely be the cause of death of pig #14.

So far, ASF-LAV strains are the most promising vaccine candidates available to induce protective immunity against the highly virulent homologous ASFV strains. To generate protective immunity, the LAV strains have to replicate just enough to induce sufficient immunity but not too much, as to overwhelm the host immune system and cause a cytokine storm that can lead to the death of the animal [52]. A recent study showed that the outcome of an ASF infection can depend on the baseline immunological and hygienic status of the pigs [53]. Pigs with higher baseline innate immune activity appear to promote immunopathological cytokine responses. Farmed animals will be of different immunological and hygienic statuses, which could be the explanation for the fatal outcomes observed following ASFV-LAV vaccination in some recipient pigs in the field. The pathogenesis associated with the mucosal versus parenteral entry of ASF-LAVs remains poorly understood. In this study, both pigs that succumbed to ASF were exposed to ASFV-GUS-Vietnam though the mucosal route. Therefore, one can argue that the pigs are susceptible to clinical disease when they are exposed to low doses of ASF-LAV strains over a longer period of time through the mucosal route. This warrants additional studies on the effect of low-dose exposure to LAV strains though the mucosal route.

## 5. Conclusions

In this study, we characterized a field ASFV isolate, ASFV-GUS-Vietnam, that lacks six genes in the MGF region. The genetic makeup of the virus closely resembles the previously described ASFV-G-ΔMGF strain; however, the origin of this virus is unclear. It was isolated from a clinical sample collected before any vaccines were approved in Vietnam. Thus, it could be a recombinant live-attenuated ASFV used as an experimental vaccine at that time. All pigs inoculated directly or that were exposed through direct contact to ASFV-GUS-Vietnam developed viremia and shed the virus for a long period of time in their oral and nasal secretions as well as their feces. Two pigs that were in direct contact with IM inoculated pigs became infected, confirming that ASFV-GUS-Vietnam can be horizontally transmitted. Two of the pigs that were exposed to ASFV-GUS-Vietnam through the ON route died of ASF, suggesting a possible increased susceptibility of pigs to ASF disease when exposed to ASFV-GUS-Vietnam via the ON route. ASFV-GUS-Vietnam, as previously reported for ASFV-G-ΔMGF, induced complete protection against a homologous challenge. However, this study raises some safety concerns associated with ASF live-attenuated recombinant viruses that could be used as vaccines and highlights the need for oversight and close monitoring for live-attenuated recombinant viruses that are being used in the field as potential vaccines to control ASF outbreaks. At least during initial field studies, the animals inoculated with live-attenuated viruses should be tested for viral shedding, along with contact animals to assess horizontal transmission. Sick and dead animals should be investigated for any potential ASF LAV reversion to virulence. This can be achieved through imposing strict surveillance in the areas where experimental ASF-LAV vaccines are being tested. Surveillance can be conducted by obtaining individual and/or easy-to-collect non-invasive samples, such as oral fluids [40]. In both experiments conducted in this study, the ASFV-GUS-Vietnam genome was detected in oral fluids on 7 dpi, a day after viremia was observed in the pens, and thereafter intermittently throughout the study, coinciding with the viremia. To differentiate the viremia and shedding caused by ASF LAV strains, molecular assays can be used [54,55,56]. Horizontal transmission, which can result through exposure of pigs to experimental ASF LAV strains via the mucosal route, can be minimized by vaccinating all of the animals in each pen. Further, off-label use of vaccines should be avoided by restricting access to experimental ASF-LAV based vaccines only to the responsible authorities. This will prevent the off-label use (improper dosages, age groups, etc.) of experimental ASF-LAV vaccines, which can lead to unwanted fatalities and the release of large amounts of ASF LAV strains into the environment.

## Figures and Tables

**Figure 1 viruses-16-00571-f001:**
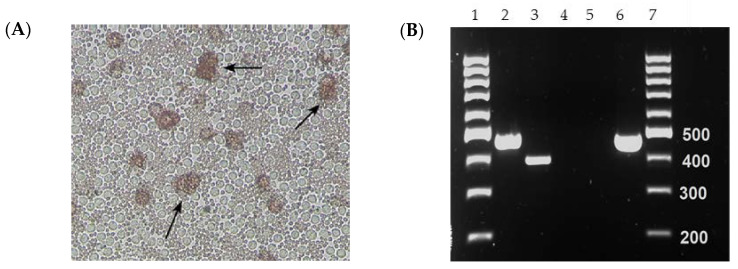
Initial characterization of the ASFV-GUS-Vietnam field isolate. (**A**) Hemadsorption observed (arrows) in porcine primary alveolar macrophage (PAM) cultures. (**B**) Confirmation of GUS insertion in ASFV-GUS-Vietnam isolated from a serum sample. The PCR amplicons were resolved by gel electrophoresis (1%). Lane 1 and 7: 100 bp Ladder. Lane 2: ASFV-GUS-Vietnam DNA. Lane 3: ASFV Georgia 2007/1 DNA. Lane 4: extraction control. Lane 5; no template control. Lane 6: *E. coli* DNA (positive control).

**Figure 2 viruses-16-00571-f002:**
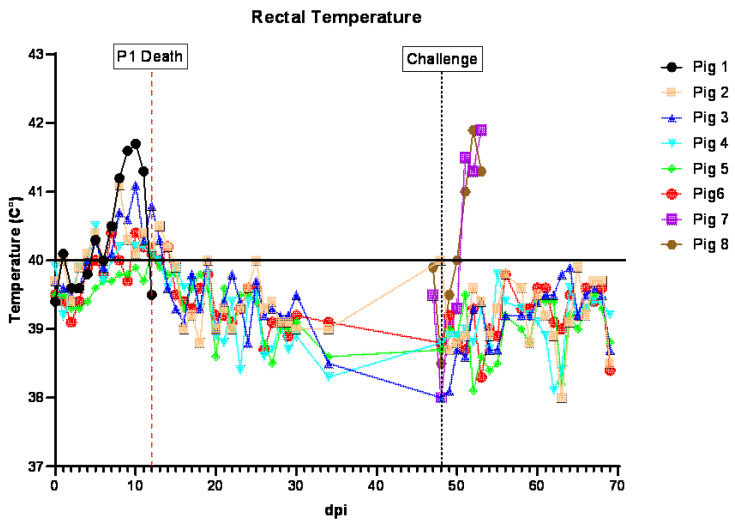
Rectal temperatures of pigs inoculated ON with ASFV-GUS-Vietnam (2 × 10^5^ TCID_50_/pig) and challenged IM with ASFV Georgia 2007/1 IM (1 × 10^3^ TCID_50_/pig). dpi = days post-infection.

**Figure 3 viruses-16-00571-f003:**
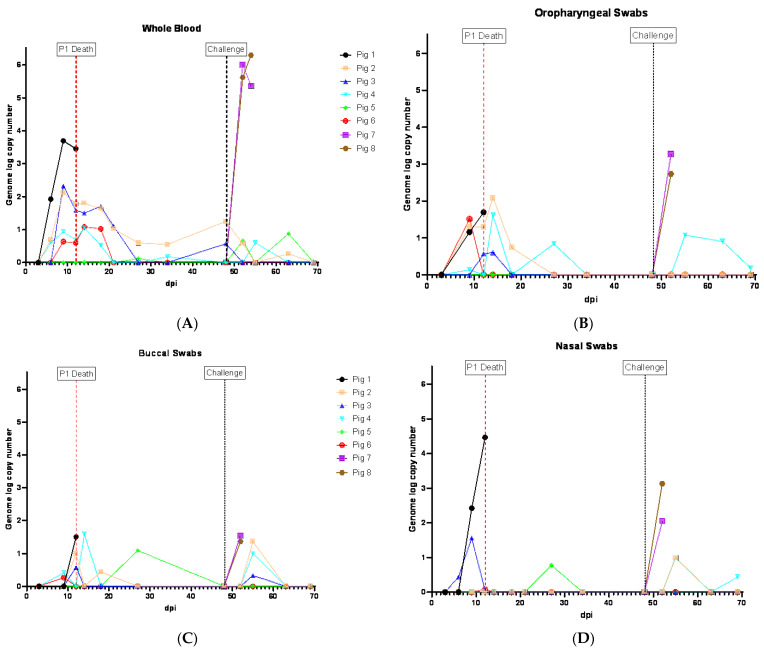
ASFV genome detection in clinical samples from pigs inoculated (ON) with ASFV-GUS-Vietnam (2 × 10^5^ TCID_50_/pig) and challenged (IM) with ASFV Georgia 2007/1 (1 × 10^3^ TCID_50_/pig). (**A**) Whole blood, (**B**) oropharyngeal swabs (OPSWs), (**C**) buccal swabs (BSWs), (**D**) nasal swabs (NSWs), (**E**) rectal swabs (RSWs), and (**F**) oral fluids. dpi = days post-infection.

**Figure 4 viruses-16-00571-f004:**
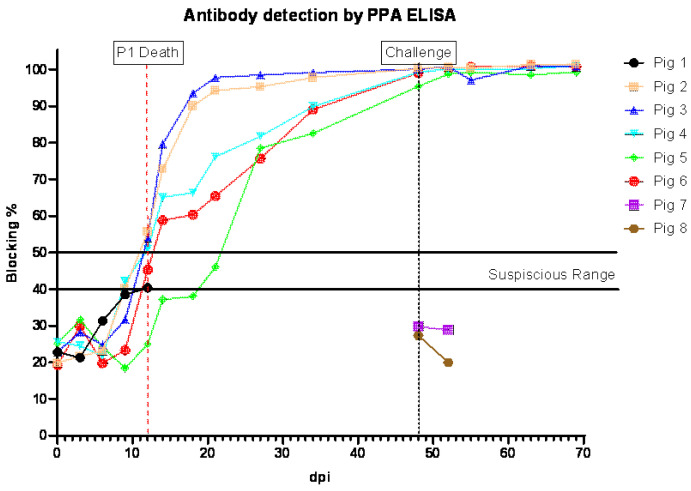
Detection of antibodies in sera collected from pigs inoculated ON with ASFV-GUS-Vietnam (2 × 10^5^ TCID_50_/pig) and challenged IM with ASFV Georgia 2007/1 (1 × 10^3^ TCID_50_/pig). dpi = days post-infection.

**Figure 5 viruses-16-00571-f005:**
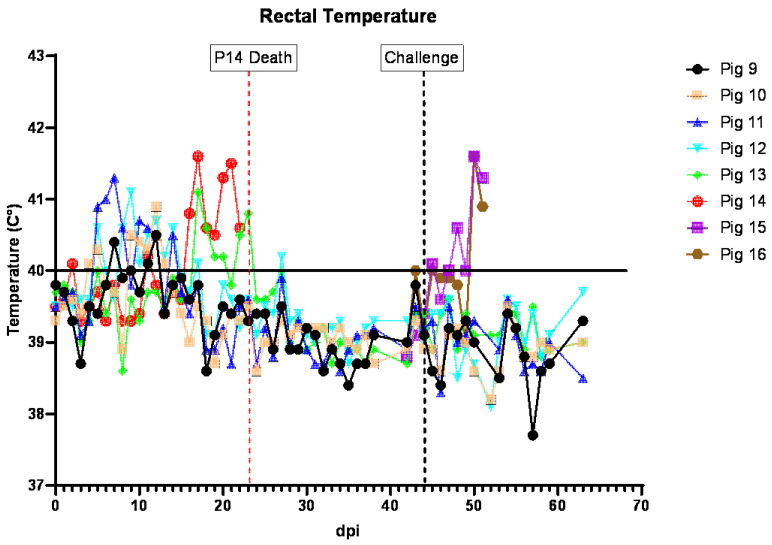
Rectal temperatures of pigs inoculated IM with ASFV-GUS-Vietnam (2 × 10^3^ TCID_50_/pig) and challenged ON with ASFV Georgia 2007/1 (2 × 10^5^ TCID_50_/pig). dpi = days post-infection.

**Figure 6 viruses-16-00571-f006:**
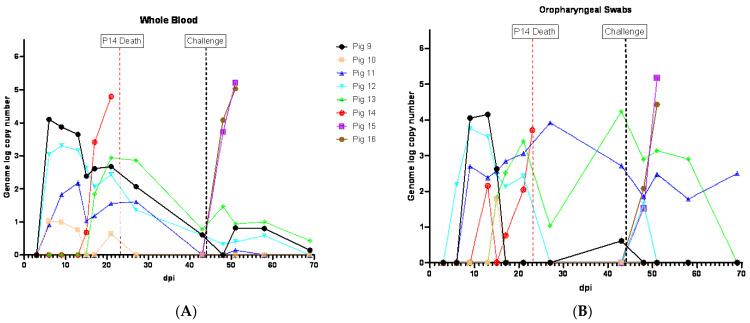
ASFV genome detection in clinical samples from pigs inoculated IM with ASFV-GUS-Vietnam (2 × 10^3^ TCID_50_/pig) and challenged ON with ASFV Georgia 2007/1 (2 × 10^5^ TCID_50_/pig): (**A**) whole blood, (**B**) oropharyngeal swabs (OPSWs), (**C**) buccal swabs (BSWs), (**D**) nasal swabs (NSWs), (**E**) rectal swabs (RSWs), and (**F**) oral fluids. dpi = days post-infection.

**Figure 7 viruses-16-00571-f007:**
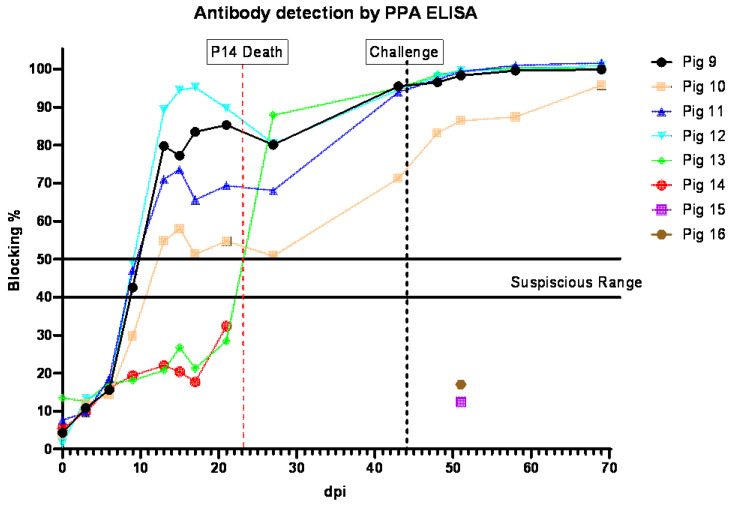
Detection of antibodies in sera collected from pigs inoculated IM with ASFV-GUS-Vietnam (2 × 10^3^ TCID_50_/pig) and challenged ON with ASFV Georgia 2007/1 (2 × 10^5^ TCID_50_/pig). dpi = days post-infection.

**Figure 8 viruses-16-00571-f008:**
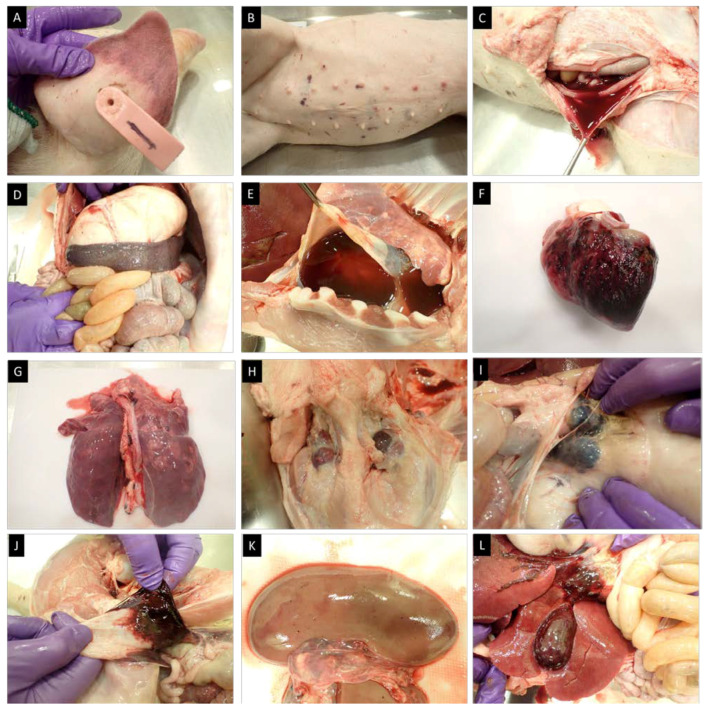
Gross pathology observed in pigs #1 and 14. (**A**) Cyanosis in the ear tip (pig #1), (**B**) ecchymotic hemorrhages in the skin (pig #1), (**C**) serosanguineous peritoneal fluid (pig #14), (**D**) enlarged spleen (pig #14), (**E**) serosanguineous pericardial effusion (pig #1), (**F**) epicardial hemorrhages (pig #1), (**G**) consolidated lungs (pig #1), (**H**) hemorrhagic submandibular lymph nodes (pig #14), (**I**) hemorrhagic gastro-hepatic lymph nodes (pig #1), (**J**) hemorrhages in the bladder (pig #1), (**K**) petechial hemorrhages in the renal cortex (pig #14), and (**L**) hemorrhagic gall bladder (pig #14).

**Figure 9 viruses-16-00571-f009:**
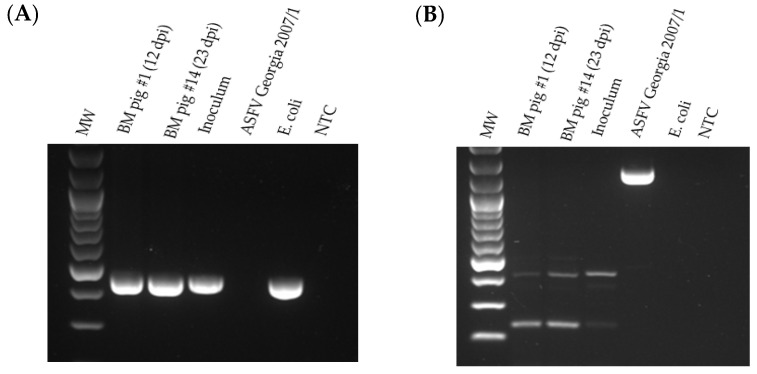
Detection of GUS (**A**) and MGF 505-2R (**B**) in the ASFV-GUS-Vietnam inoculum and in bone marrow (BM) samples collected from pigs #1 and #14 that died of ASF. NTC = no template control. MW = 100 bp ladder. *E. coli* = *E. coli* DNA.

**Table 1 viruses-16-00571-t001:** Detection of 360-14L gene by real-time PCR in the ASFV-GUS-Vietnam inoculum and in bone marrow (BM) collected from pigs #1 and #14 that died of ASF. EC = extraction control. NTC = no template control. Tignon RT-PCR detected the p72 gene that is present in both ASFV Georgia 2007/1 and ASFV-GUS-Vietnam. ASFV MGF 360-14L is present only in ASFV Georgia 2007/1.

Sample	MGF 360-14L	Tignon	ß-Actin
Bone Marrow (pig #1)	-	24.49	23.72
Bone Marrow (pig #14)	-	21.32	22.65
ASFV-GUS Inoculum	-	23.68	30.67
ASF Georgia 2007/1	25.26	25.72	29.63
EC	-	-	-
NTC	-	-	-

**Table 2 viruses-16-00571-t002:** All nucleotide changes observed in the whole genomes of ASFV-GUS-Vietnam inoculum (inoculum), and the virus in tissues from pig #1 (ASFV-GUS-Vietnam-1) and pig #14 (ASFV-GUS-Vietnam-14), compared to that in the ASFV Georgia 2007/1 whole genome sequence. Rows highlighted in gray show a significant mutation and their effects. * Position relative to ASFV Georgia 2007/1. NSM = non-synonymous mutation.

Position *	Gene	Georgia 2007/1	Inoculum	Pig #1	Pig #14	Effect
1371	5′ ITR	-	A	A	A	No effect
1390	Non-coding	C	-	-	-	No effect
7013	MGF-110-1L	-	G	G	G	Altered reading frame, extended CD
14,235	MGF-110-10-L—MGF110-14L fusion	CCC	---	---	---	Deletion of one aa
17,110	MGF 360-4L	G	G	R (66.0% G, 33.2% A)	R (66.0% G, 33.2% A)	If A allele: A(Ala) > V(Val) (NSM)
17,633	Non-coding	-	G	G	G	No effect
17,847	Non-coding	--	GG	GG	GG	No effect
19,799	Non-coding	G	-	-	-	No effect
20,006	ASFV G ACD 00350 (poly G region gene)	GGG	---	---	---	Deletion of one aa
21,806	Non-coding	-	G	G	G	No effect
27,428	Non-coding	T	-	-	-	No effect
42,767	MGF 505-7R	C	C	Y (71.4% C, 26.7% T)	C	If T allele: No amino acid change
46,691	MGF 505-10R	G	A	A	A	NSM: E(Glu) > K(Lys)
59,528	F778R	C	C	Y (73.6% C, 23.4% T)	C	If T allele: No amino acid change
98,378	B438L	A	G	G	G	No Amino acid change
135,282	NP419L	A	A	R (64.8% A, 33.0% G)	A	If G allele: V(Val) > A(Ala)
153,855	H124R	G	G	R (68.3% G, 29.7% A)	G	If A allele: No amino acid change
155,799	H233R	T	T	C	C	V(Val) > A(Ala): NSM:
167,188	E199L	C	G	G	G	A(ALA) > P(Pro): NSM
179,360	MGF 505-11L	A	A	T	T	L(Leu) > H(His): NSM
190,234	3′ ITR	A	A	A	R (65.9% A, 34.1% G)	

## Data Availability

All data related to this study will be made available upon request.

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
