# Peer review of "Characterization of an African Swine Fever Virus Field Isolate from Vietnam with Deletions in the Left Variable Multigene Family Region"

_viruses, 2024, doi:10.3390/v16040571_

Round 1

Reviewer 1 Report

Comments and Suggestions for Authors

“ASFV-GUS- Vietnam field isolate is able to induce complete protection in the majority of the pigs against highly-virulent homolugus ASFV challenge…”

These are solutions out of an emergency situation and shows how important an adequate vaccine would be.

Several precarious points in the infectious processes should be focused and some thoughts about that,

“The infected animals shed the virus in all body secretions and excretions, and transmit it to naïve contact pigs.”

Some pigs are certainly naïve, others have antibodies that are not protective.

“none of the pigs developed clinical signs. They did however, continue to shed virus intermittently”

Difficult, some are shedding, others are not, no homogeneous infection groups, not comparable to a real vaccination.

In other words, you don't necessarily know your stand. The flow and the incidence of infection is difficult to assess.

Author Response

Response to reviewer’s comments – Viruses-2923252

Reviewer 1

Comments and Suggestions for Authors

“ASFV-GUS- Vietnam field isolate is able to induce complete protection in the majority of the pigs against highly-virulent homolugus ASFV challenge…”

These are solutions out of an emergency situation and shows how important an adequate vaccine would be.  Several precarious points in the infectious processes should be focused and some thoughts about that,

“The infected animals shed the virus in all body secretions and excretions, and transmit it to naïve contact pigs.”  Some pigs are certainly naïve, others have antibodies that are not protective.

 “none of the pigs developed clinical signs. They did however, continue to shed virus intermittently”. Difficult, some are shedding, others are not, no homogeneous infection groups, not comparable to a real vaccination.

In other words, you don't necessarily know your stand. The flow and the incidence of infection is difficult to assess.

Response: We thank the viewer for the insightful comments. We modified the sentences to reflect the comments from the reviewer.

The sentence “The infected animals shed the virus in all body secretions and excretions, and transmit it to naïve contact pigs.” now says, “The infected animals shed the virus in body secretions and excretions, and transmit it to contact pigs”. Lines 76-77.

The sentence “ “none of the pigs developed clinical signs. They did however, continue to shed virus intermittently” Now says “ However, some animals continued to shed virus intermittently”. Lines 615-616.

Reviewer 2 Report

Comments and Suggestions for Authors

In this manuscript the authors collected serum samples from family farms in North Vietnam that had experienced ASF outbreaks between 2019 and 2021. These were tested for ASFV genome and for the presence of the b-GUS cassette, a marker used in the generation of some gene-deleted ASFV viruses. Whole genome sequencing of an original isolate and two viruses isolated from experimental pig infections was performed. One isolate, ASFV-GUS-Vietnam, was used to infect pigs by the oral nasal route (2 X 105) and monitor clinical signs, viremia, virus shedding and macroscopic lesions typical of ASF. One pig reached the humane endpoint. At 48dpi surviving pigs were challenged in parallel with two control pigs in a separate room. In the methods it is described the challenge was with ASFV-GUS-Vietnam whereas in the results section (line 320 ) it states that Georgia 2007/1 was used for challenge which is I presume correct. All surviving pigs survived challenge and were euthanised 21 days after this second challenge. In a second experiment 4 of 6 pigs in pen were immunised by the IM route with ASFV-GUS-Vietnam (2X10 3) and clinical signs and sample analysis carried out as for the first experiment. The results showed that most pigs developed moderate clinical signs including high temperatures after immunisation and one reached a humane endpoint. The remaining  inoculated pigs survived challenge with Georgia 2007/1 whereas the two pigs in direct contact reached the humane endpoint. Whole  genome sequencing showed an insertion of the GUS cassette at the site of an  identical deletion of 6 MGF genes to that described by O’Donnell et al., 2015. Additional small genome changes were detected. In the current study greater clinical signs were observed following immunisation of pigs with this virus than in previously published studies. This point is discussed by the authors.

The finding of genetically modified ASFV very similar to a previously described gene-deleted ASFV virus from farms in N. Vietnam is of great interest. The origins of this virus are unknown and of note the virus was isolated prior to the current vaccine trials in Vietnam.

Specific points:

1.      It is important to understand the prevalence of the ASFV-GUS-Vietnam virus in Vietnam. How many samples analysed, prevalence of ASFV alone and ASFV GUS positive samples?

2.      In results section 3.1 it should be more clearly explained the expected sizes of the PCR fragments and their origin. Bands are detected in both lanes 2 and 3 (ASFV-GUS-Vietnam and Georgia 2007/1 DNA respectively). Since the Georgia 2007/1 genome should not contain the GUS gene I presume the primers used are ASFV specific. However the difference in size of the fragments is quite small and couldn’t contain the entire GUS fragment?  The significance of the Sanger sequencing of the amplicon showing a match to sus scrofa genomic DNA from chromosome 17 is also not explained.

3.      Clarify which virus was used for challenge at 48 dpi (ASFV-GUS-Vietnam or Georgia 2007/1).

4.      A detailed description of clinical signs observed is given (lines 284 to 331, 383 to 409) and a figure showing rectal temperatures (fig 2). It would be useful if the authors included a previously used clinical scoring system (eg King et al., or Gallardo et al) and a figure showing these to better visualise the results.

5.      Line 543 the significance of absence of amplicons from the MGF505-2R or MGF-260-14L  is not explained.

Author Response

Responses to Specific points:

  1. It is important to understand the prevalence of the ASFV-GUS-Vietnam virus in Vietnam. How many samples analysed, prevalence of ASFV alone and ASFV GUS positive samples?

This study was based on two ASFV genome and GUS positive samples detected when ~ 414 serum samples were screened. This information was added the Results (Line 269) and the Discussion (Line  617). We continue to screen ASFV positive whole blood, tissue and serum samples collected in Hanoi, Vietnam for GUS gene and the findings from the ongoing studies will be shared in a future publication.

  1. In results section 3.1 it should be more clearly explained the expected sizes of the PCR fragments and their origin. Bands are detected in both lanes 2 and 3 (ASFV-GUS-Vietnam and Georgia 2007/1 DNA respectively). Since the Georgia 2007/1 genome should not contain the GUS gene I presume the primers used are ASFV specific. However the difference in size of the fragments is quite small and couldn’t contain the entire GUS fragment?  The significance of the Sanger sequencing of the amplicon showing a match to sus scrofa genomic DNA from chromosome 17 is also not explained.

      The primers used in the conventional PCR mentioned under section 3.1 were GUS gene specific as described by O'Donnell et l., 2015 (reference 42 in the manuscript). The primers amplified a part (471 bp) of the GUS gene as expected when ASFV-GUS-Vietnam DNA (propagated in PPLs) and E. coli DNA (positive control) was used. However when we used DNA extracted from ASFV Georgia 2007/1 propagated in PPLs, unexpectedly we saw an amplicon that was smaller than the GUS amplicon. Sanger sequencing was used to confirm that the smaller amplicon was from the pig genome, and is therefore non-specific amplification that happened when the specific target (GUS) was not available. We revised section in 3.1. to clarify this further. 

  1. Clarify which virus was used for challenge at 48 dpi (ASFV-GUS-Vietnam or Georgia 2007/1).

In both experiments, for challenge ASFV Georgia 2007/1 was used.

  1. A detailed description of clinical signs observed is given (lines 284 to 331, 383 to 409) and a figure showing rectal temperatures (fig 2). It would be useful if the authors included a previously used clinical scoring system (eg. King et al., or Gallardo et al) and a figure showing these to better visualise the results.

      We thank the reviewer for this suggestion, and this information was included in the manuscript (Lines 335-340) and Lines 447-451) and a supplementary figure (Figure S1) was added

  1. Line 543 the significance of absence of amplicons from the MGF505-2R or MGF-260-14L is not explained.

The sentenced was modified (Lines  558-561) as follows - No MGF 505-2R amplicon was detected in the ASFV-GUS-Vietnam inoculum or in bone marrow samples from pig #1 and #14, suggesting that wild-type (ASFV genotype II epidemic strain circulating in Vietnam) ASFV was not responsible for the fatal outcome observed in these two pigs.

Reviewer 3 Report

Comments and Suggestions for Authors

This is a very typical study to investigate ASFV pathogenesis in pigs. In this study, a filed LAV was isolated in Vietnam and characterized in pigs using mucosal, intramuscular infections followed by virulent Georgia/07 challenge to see if any protective effect is there. In the end this study highlights the concern of illegal administration of lab LAVs which are typically not fully characterized.

I did not understand the GUS insertion after reading the first paragraph of discussion. So a bit details of the gene in recombinant ASFV construction should be introduced in the very beginning and briefly mentioned in results. Similarly, the pathogenesis of MGF 6 gene deletion vaccine, as well as reports of illegal use of lab vaccine can be described briefly in introduction to help understand the whole story.

Author Response

Response: We thank the reviewer for the valuable comments and feedback.

  1. We have added a sentence briefly describing the genetic makeup of the virus into the Introduction. Lines 87-88.
  2. Pathogenesis of MGF 6 gene deletion vaccine has been described in the discussion. Lines 638 -646, and 672 and 678.
  3. Potential illegal use of ASFV vaccine candidates in the field has been mentioned in the introduction. Line 64-65.